# Ventilation-Based Strategy to Manage Intraoperative Aerosol Viral Transmission in the Era of SARS-CoV-2

**DOI:** 10.3390/life14030313

**Published:** 2024-02-28

**Authors:** Ayoola T. Brimmo, Ayoub Glia, Juan S. Barajas-Gamboa, Carlos Abril, John Rodríguez, Matthew Kroh, Mohammad A. Qasaimeh

**Affiliations:** 1Division of Engineering, New York University Abu Dhabi, Abu Dhabi 129188, United Arab Emirates; tab478@nyu.edu (A.T.B.); ag6042@nyu.edu (A.G.); 2Department of Mechanical and Aerospace Engineering, New York University, New York, NY 11201, USA; 3Digestive Disease Institute, Cleveland Clinic Abu Dhabi, Abu Dhabi 112412, United Arab Emirates; barajaj@clevelandclinicabudhabi.ae (J.S.B.-G.); abrilc@clevelandclinicabudhabi.ae (C.A.); rodrigj2@clevelandclinicabudhabi.ae (J.R.); krohm@ccf.org (M.K.); 4Digestive Disease and Surgery Institute, Cleveland Clinic, Cleveland, OH 44195, USA; 5Department of Biomedical Engineering, New York University, New York, NY 11201, USA

**Keywords:** SARS-CoV-2, COVID-19, operating room, ventilation systems, safety, aerosol viral transmission

## Abstract

In operating theaters, ventilation systems are designed to protect the patient from airborne contamination for minimizing risks of surgical site infections (SSIs). Ventilation systems often produce an airflow pattern that continuously pushes air out of the area surrounding the operating table, and hence reduces the resident time of airborne pathogen-carrying particles at the patient’s location. As a result, patient-released airborne particles due to the use of powered tools, such as surgical smoke and insufflated CO_2_, typically circulate within the room. This circulation exposes the surgical team to airborne infection—especially when operating on a patient with infectious diseases, including COVID-19. This study examined the flow pattern of functional ventilation configurations in view of developing ventilation-based strategies to protect both the patient and the surgical team from aerosolized infections. A favorable design that minimized particle circulation was deduced using experimentally validated numerical models. The parameters adapted to quantify circulation of airborne particles were particles’ half-life and elevation. The results show that the footprint of the outlet ducts and resulting flow pattern are important parameters for minimizing particle circulation. Overall, this study presents a modular framework for optimizing the ventilation systems that permits a switch in operation configuration to suit different operating procedures.

## 1. Introduction

The use of specialized ventilation systems (VSs) in operating rooms (ORs) dates back to 1946, when Bourdillon and Colebrook reported the effect of their VS in attaining “safe levels” of airborne pathogen-carrying particles [1]. The system functioned based on the “air flow piston effect”, where a laminar layer of clean air is introduced from the ceiling to continuously push the contaminated air downwards and away from the surgical table [2]. Over the years, this ventilation model has proven to be efficient in curtailing post-operative sepsis [3,4,5]. While its design and implementation have gone through several modifications during this period, the same operating principle has always guided the commissioning of VSs in newly built operating theaters [6,7,8]. Since the movement of the patient during surgery is minimal, it was not conventionally considered as a significant contamination risk to the surgical team [9]. As such, the primary target of these VS setups has been focused on protecting the patient’s open incisions from airborne contaminants produced by the surgical team [10], not vice versa.

In modern operating room air distribution design, primary considerations revolve around maintaining a sterile surgical zone and integration with medical equipment. On one hand, the objective is to establish and uphold a sterile surgical zone encompassing both the patient and the surgical team. On the other, the design must seamlessly coordinate with sophisticated medical imaging devices, task lighting, medical gas columns, and other ceiling-mounted equipment, ensuring optimal functionality within the operating room. The guidelines provided by the FGI (Facility Guidelines Institute) for the Construction of Health Care Facilities and the ASHRAE 170-2017 Standard for the Ventilation of Health Care Facilities delineate specific airflow requirements for operating rooms. For instance, the room should maintain positive pressurization, with a minimum of 20 total air changes per hour (ACH). Additionally, the airflow, filtered appropriately, should follow a unidirectional downward pattern through a laminar array designed for 25 to 35 cfm/sq. ft. Furthermore, a primary supply diffuser array must cover at least 70% of the surgical zone, defined by the footprint of the operating table and a 12-inch perimeter offset. In the context of traditional operating rooms, there is typically ample space to accommodate a large laminar diffuser array directly above the operating table. This spatial arrangement facilitates the straightforward fulfillment of the specified ASHRAE 170 criteria when devising a ventilation and ceiling system for optimal operating room functionality [7].

In recent times, the principles of VSs have been challenged, as the use of powered surgical tools releases surgical aerosols and insufflated CO_2_ during minimally invasive procedures, which increases the exposure risk of the surgical team to potentially harmful pathogens and particles [11,12]. Previous evidence has identified the presence of viruses in surgical aerosols, including human immunodeficiency virus [13], polio virus [14], human papilloma virus [15], and hepatitis B virus [16]. In one of these studies, the hepatitis virus was found to be present in surgical aerosols produced while operating on 10 out of 11 positively tested patients [16]. Other studies have also reported transmission of such viruses to the surgical team [17,18,19], which highlights the risk posed to the surgical team. These earlier studies gained a lot of importance at the beginning of the current pandemic. Surgical societies and international healthcare organizations established a consensus to develop guidelines and protocols to manage surgical procedures; however, this proved insufficient for managing the risks, as more than 28 million elective surgeries were cancelled or postponed worldwide, compromising essential medical care to patients [20].

While there are limited data regarding intraoperative aerosol viral transmission (IAVT) of the severe acute respiratory syndrome coronavirus 2 (SARS-CoV-2), initial reports on infected patients who underwent surgery revealed the presence of the virus in samples of intra-abdominal tissues, peritoneal fluid, and stool [20,21,22]. These clinical findings and the high infection rates of the SARS-CoV-2 prompted the medical community to take measures to prevent hospital-acquired infections, especially when operating on positive SARS-CoV-2 patients.

Safe evacuation of insufflated CO_2_ and surgical aerosols during laparoscopic and robotic surgery can be achieved using desufflators [23,24], smoke evacuators [25], and electrostatic precipitation tools [26,27]. However, these devices were designed to improve the surgeon’s visibility and minimize exposure to fumes, not for the evacuation of CO_2_. In addition, none of these devices have been tested with viruses [28]. In recent times, aerosol encapsulating boxes, inspired by the pediatric head box experience, have been adapted to lend protection against aerosol exposure of the surgical team [29,30]. However, aerosol boxes have raised some criticism concerning their obstruction of accurate manipulation of surgical devices [30]. Optimization of the OR’s VS to favorably minimize the circulation of surgical smoke and aerosols is a hypothetical option to minimize aerosol exposure without obstructing the standard operating procedures. This process requires assessments of the airflow pattern and airborne particle concentrations with different ventilation configurations. In the past three decades, several studies have contributed to the development of effective guidelines for controlling airborne infections in healthcare settings [28,29]. One example of such guidelines is the requirement to maintain a minimum air change rate of 20 per hour in the operating room (OR) to minimize the deposition of airborne contaminants in the surgical area and reduce the likelihood of surgical patients contracting surgical site infections [30]. In another relevant study, an overview of models and algorithms utilized in computational fluid dynamics (CFD) simulations for pathogen transmission research is provided [31]. This work highlights the representation of pathogens as particles or gases, integrates epidemic models for enhanced accuracy, and emphasizes the crucial role of airflow patterns in pathogen transmission. Additionally, the study introduces advanced CFD methods such as the Lattice Boltzmann Method (LBM), Porous Media models, and Web-based approaches, offering a diverse set of tools for intricate investigations in this field [31]. More particularly, in a recent study, CFD methods were used to simulate airflow within various types of OR setups, suggesting that air exits the space exclusively through a restricted number of outlets—specifically, two to four outlets on just two walls are specified—instead of exiting through a perforated floor, resulting in airflow turbulence in certain areas of the room and unpredictable particle movement [7,32,33,34,35,36]. A recent study by Wagner et. al. [32] performed experimental tests to evaluate air particle concentrations in ORs. The tests were conducted by releasing particles at the OR table and consequently evaluating particle concentration within the room. Their data showed that the airflow within the OR caused higher particle concentration near the OR walls in comparison to the center of the room. While this promised a safer zone for patients and surgeons, the rest of the surgical team, including surgical assistants, nurses, anesthetists, and other operators, might face higher risks. The study did not explain what risks were involved in this particle concentration near the walls and did not explain how to solve the problem. Currently, there are no studies that evaluate strategies to improve existing VS performances to reduce IAVT. Any changes or modifications to the structure in operating rooms has the potential to avert the suspension of surgical procedures during a possible future pandemic, thereby avoiding the severe consequences it could inflict on patients and the surgical team. In this regard, the structural implementation of the operating rooms would be in line with the processes of guarantees of non-repetition proclaimed by the application of Transitional Justice [37]. Based on this, the objective of this study was to examine the flow pattern of functional ventilation configurations and develop ventilation-based strategies to protect both the patient and the surgical team from IAVT. Therefore, we studied functional operating rooms (Figure 1a,b) and developed a corresponding computational fluid domain with matching geometry to study the ventilation flow further (Figure 1c,d).

## 2. Materials and Methods

The techniques employed to examine and assess the effectiveness of operating room designs concerning particle movement within and around the sterile field include (1) computational fluid dynamics modeling (CFD) and (2) airflow experimental tests using neutral buoyancy bubbles. CFD has been traditionally used to illustrate how airflow patterns within the OR can affect the induction and migration of particles. In a recent study, particle trace analysis comparing the Moving Downstream Air (MDA) design with the Single Laminar Diffuser (SLD) design revealed that SLD performed significantly better in pulling contaminants away from the surgical site under equivalent conditions. Additionally, a neutral buoyancy bubble experiment revealed the impact of low-pressure zones resulting from airflow blockage [38].

In this study, we used the commercially available multiphysics computer-based simulation package, COMSOL, to develop CFD models of a standard empty OR with varying ventilation configurations. Furthermore, experimental airflow patterns collected from an existing fully equipped OR were compared with the moder’s result for validations. The experimental airflow patterns were measured by tracing the flow of bubbles in a fully equipped OR, but without the presence of the surgical team. In view of minimizing surgical aerosols circulation, the airflow pattern produced from three different ventilation configurations were evaluated using particle half-life (PHL) and particle elevation (PE). PHL and PE are common parameters used in the field of particle physics and aerosols and hence implemented in this study as a measure of circulation potential.

### 2.1. Operating Room Settings

The experimental setup used for model validation corresponded to a real OR at a commercial medical facility in Abu Dhabi in the first quarter of 2021. The room measured 6 × 5 × 3.8 m^3^ (L × B × H) and contained standard laparoscopic OR equipment, which included an anesthesia machine, laparoscopic tower, anesthesia car and tables for surgical technicians, etc. The installed VS consisted of a square laminar air flow inlet duct positioned on the ceiling and measuring 4 × 4 m^2^ (Figure 1a). Two rectangular outlet vents measuring 0.4 × 1 m^2^ (L × H) were positioned at the ground level of the walls located in two opposite corners of the room (Figure 1b). A corresponding domain of the CFD model was created with the same size and ventilation configuration (Figure 1c). In addition, the operating table and trocar access sites were included as a fixed structure and the release outlet of the surgical aerosol, respectively.

Experimental measurements were performed using a uniform laminar air supply with a velocity of 0.18 m/s at room temperature. The outlet ducts were operated at zero pressure conditions. The release of particles from the trocars was reproduced using a bubble generator with a particle release velocity of 5 m/s. Videos of released bubbles were taken using digital single-lens reflex (DSLRS) cameras (Canon 7D) positioned at two ends of the OR and directed to face the bubble generator positioned on the operating table (see purple arrows in Figure 1d). Upon reaching a steady operation of the VS within the OR, bubbles were released to visualize the flow pattern of particles during the release of insufflated CO_2_.

### 2.2. CFD Model

The CFD model was developed as a steady state, three-dimensional, and isothermal flow field based on the Navier–Stokes equations:(1)$$\rho \left(\vec{u}\cdot \nabla \right)\vec{u}=\nabla p+\mu \cdot \nabla^{2}\vec{u}$$
where ρ is the fluid’s density, $\vec{u}\;$is the velocity vector in the cartesian coordinates, and μ is the fluid’s dynamic viscosity. The left-hand side of the equation represents the flow force, while the first term on the right-hand side represents the generated pressure gradient and the second term represents the viscous forces. The airflow pattern was generated by operating the VS with the inlet duct at a velocity of 0.18 m/s, the outlet duct at zero pressure conditions, and particle release from a 2 cm trocar at 5 m/s. Turbulence is accounted for by the *k*-*ε* turbulence model, which introduces the turbulence kinetic energy (*k*) and dissipation rate (*ε*) as dependent variables to account for turbulence with the viscosity term [31].
(2)$$\mu_{T}=\rho c_{\mu }\frac{k^{2}}{\varepsilon }$$
where c is a model constant and the transport equations for *k* and ε are presented in Equations (3) and (4), respectively:(3)$$\rho \frac{\partial k}{\partial t}+\rho u\cdot \nabla k=\nabla \cdot \left(\left(\mu +\frac{\mu_{T}}{\sigma_{k}}\right)\nabla k\right)-\rho \varepsilon$$
(4)$$\rho \frac{\partial k}{\partial t}+\rho u\cdot \nabla \varepsilon =\nabla \cdot \left(\left(\mu +\frac{\mu_{T}}{\sigma_{\varepsilon }}\right)\nabla \varepsilon \right)-c_{\varepsilon }\rho \frac{\varepsilon^{2}}{k}$$

The trajectory of particles released by the trocars were computed using the Particle Tracing Module, which is governed by the bidirectional coupled particle tracing model. The model is governed by the momentum conservation principle according to the equation:(5)$$\frac{d(m_{p}v)}{dt}=F_{p}$$
where $m_{p}$ and v are the particle mass and velocity, respectively, and $F_{p}$ is the external force vector. The entire domain was bounded by fixed walls while air flow conditions were defined based on the measured velocity of the inlet flow and the pressure of the outlet flow. For the particle tracing computation, the wall boundaries were defined to allow for particle bouncing while the outlet ducts were defined to freeze particle motion. A total of 161,423 domain elements, 8996 boundary elements, and 604 edge elements were generated for the entire mesh, with a fine grid adaptation concentrated around the trocar location. Computation was performed using the Jacobi iterative model with a Generalized Minimal Residual Method (GMRES) solver and residual tolerance of 0.01—a combination of the Jacobi model [32] that provides a simple and memory-efficient solver based on GMRES methods [39] for solving the general linear system of the form *Ax* = *b*.

### 2.3. Model Validation

The view of the camera positioned at the corner with an outlet vent duct was arbitrarily tagged as View-plane 1 (Figure 2a), while the view of the camera positioned at the corner without an outlet vent duct was tagged as View-plane 2 (Figure 2b). During the experiments, the focus area was defined with the trocar centralized within the picture frame. However, due to the limited camera field of view, this area was limited to the highlighted dotted square area shown in Figure 2c,d. Real-time particle trajectory videos were collected on this highlighted region from both View-planes, and the trajectory of released bubbles was manually traced from frame to frame using the freely available image post-processing tool, Image J (Figure 2e–h).

### 2.4. VS Configurations

The first configuration (C1) mimicked the existing system within the OR where the experimental tracings were performed (Figure 3a)—an OR with a square inlet duct on the ceiling and two rectangular outlet ducts in opposite corners. The second configuration (C2) mimicked an existing OR VS configuration located in Ohio, USA (Appendix A), while the third configuration was deduced based on the simulation results of the other two configurations and proposed to reduce PHL and PE by adding two more outlet ducts in corners that experienced high circulation of particles. While the second variation adapted the concentric square inlet configuration and extended the length of the rectangular outlet ducts (Figure 3c), the third variation (C3) adapted the single square inlet duct and four outlet ducts in all corners of the room (Figure 3e).

### 2.5. Measurements

All three configurations were compared based on the adapted quantification of circulation—particle half-life (PHL, time required for half of the released particles to stagnate) and particle elevation (PE, the maximum height a released particle can reach). In concurrence with recent virus transmissibility-related studies [40,41], these parameters were selected based on the suitability of PHL and PE to represent exposure time and distance from inhaling proximity, respectively. To enable this quantification, the particle tracing models were initiated by releasing 1000 particles from the trocar during the first 5 s of the CFD simulation. Surface probes were then used to count the total particles reaching zero velocity on the ground level or exiting via the outlet ducts.

## 3. Results

During the validation experiments, the flow profile portrayed an upward trajectory of the particles immediately after release, which was eventually pushed downwards by the directly imposing laminar airflow from the inlet duct on the ceiling. This was followed by a symmetrical split of the particles towards the corners of the rooms without an outlet air duct and then an upward circulation of the flow profile towards the outlet ducts. Correspondingly, particle tracings on the focus area of View-plane 1 revealed a high density of symmetrical flow split (Figure 2e), while View-plane 2 portrayed minimal flow split (Figure 2f). Overall, this trend was in concurrence with the CFD calculations (see Appendix A). This was highlighted by high density of symmetrical flow split and low density of flow recirculation within the focus area of View-plane 1, which are shown with the dotted square in Figure 2c. In addition, this trend was confirmed by the low density of symmetrical flow split and high density of flow recirculation within the focus regions of View-plane 2, which are shown with the dotted square in Figure 2d. Tracings of the CFD-calculated particle trajectory within the focus area also showed significant symmetrical flow split on View-plane 1 (Figure 2g) and minimal flow split on View-plane 2 (Figure 2h). In Figure 2f, the experimentally measured particle trajectories do not show circulation in correspondence to numerically calculated results of the same view plane (Figure 2d,h). This can be attributed to the obstruction of camera’s view of View-plane 2 by bubbles flowing closer to the camera at that corner. Nevertheless, similarities in flow-split characteristics between the experimentally measured and numerically modeled flow pattern results suggest an acceptable fitness of the numerical model. The validated numerical model was then used to compare three different ventilation configurations proposed for the same OR.

In Configuration C1 (Figure 3a), adapted from an existing OR design, the maximum PE is 3.43 m, while the PHL is 217 s. Using this configuration, particles attain their maximum height and show a significant degree of circulation in the room corners that have no outlet ducts (see Figure 3b and Appendix A). In Configuration C2, also adapted from an existing OR design, the concerted square configuration is adapted for the inlet ducts, and the outlet ducts are extended across the walls of the room (See Figure 3c). The concerted square inlet configuration was adapted from a functional OR, as shown in Appendix A. This results in a PHL of 413 s and a maximum PE of 3.75 m. The significant increase in PHL is observed to occur as a result of repeated circulation within the patient’s location (see Figure 3d and Appendix A). The concerted square inlet duct configuration effectively creates a stagnation zone in that region and acts like a trap to the released particles, which increases the risk of surgical site infections. However, once the particles escape the region, they are immediately pulled out of the domain by the outlet ducts. Based on this result, the advantage of a square inlet duct and increased outlet duct footprint for minimizing the resident time within and outside the patient area, respectively, is validated. These findings explain the higher particle concentration near the OR’s walls reported earlier [32]. This confirms the power of combining computer simulation-based analysis with experimental tests.

As such, these advantageous features of Configurations C1 and C2 are combined to propose the new Configuration C3. In this updated configuration, we suggest employing a square outlet duct and four inlet ducts positioned in the corners of the room (refer to Figure 3e). The intention behind covering all four corners of the operating room is to minimize any negative pressure spots that might alter airflow patterns, potentially surpassing our designated particle maximum elevation threshold. This results in PHL and PE of 58 s and 1.45 m, respectively, which demonstrates a significant reduction in particle circulation (see Table 1). With the proposed C3 configuration, particles are also observed to be pushed directly to the ground level, where they stagnate on the floor due to the symmetrical splitting effect of the four outlet ducts (see Figure 3f).

Comparing the steady state flow streamlines across the center ZX-plane of each configuration, a direct relationship between PHL/PE and circulation is depicted. In this plane, while the flow streamline of configuration C1 demonstrates high-density circulation close to the walls (see Figure 4a), the flow streamline of configuration C2 demonstrates significant circulation within the patient area (see Figure 4b). In contrast, configuration C3 portrays streamlines with direct flow to the floor of the room with very minimal circulation within the plane (see Figure 4c and Appendix A). In order to substantiate the steady state particle dynamic trend observed from the flow streamline, the particle-tracing model was deployed with continuous particle release from the trocar to simulate the 3D particle distribution in the steady state. Results from that model corroborate the streamline patterns of Figure 4a–c—high density, elevation, and circulation of particles outside of the patient area in configuration C1 (see Figure 4d) and within the patient area of configuration C2 (see Figure 4e). This suggests that ORs with configuration C1 harbor a high-risk infection zone for the surgical team located far away from the operating table, while configuration C2 harbors a high-risk infection zone for both the patient and the surgical team located close to the surgical table. On the contrary, while the majority of the released particles settle within the patient’s zone in Configuration C3 (see Figure 4f), the particles attain very low elevations and hence present very little risk to either the patient or the surgical team. As such, a VS based on Configuration C3 effectively reduces the risk of surgical site infection of both the open incision and the surgical team. Furthermore, the configuration presents a modularity advantage that allows the OR to work in different modes depending on the circumstances.

## 4. Discussion

In this study, the particle tracing numerical model was developed for the configuration of an existing and functional OR, and representative particle tracing experiments were used to validate the computational model. It is important to note here that the computer numeral model simulated an empty OR, while the experimental airflow tracing tests were performed in a fully equipped OR but without the presence of the surgical team. The use of near-neutral buoyancy bubbles is not intended to mimic SARS-CoV-2-carrying aerosols, but to dynamically respond to changes in the flow field and to trace flow streamlines, allowing us to confirm the presence of the large vortices generated in the adjacent corners of the OR, as observed in the numerical model. Although the adapted experimental procedure modeled the laparoscopic surgery CO_2_ venting procedure in the absence of closed evacuation systems, similar flow profiles are expected for other release events, e.g., during trocar placement, instrument changing, and/or specimen extraction procedures. Furthermore, since the trocar port and positioning are similar, we expect this study’s result to be applicable for laparoscopic, robotic, and endoscopic surgeries. While performing direct measurements provided the most realistic airflow information (as demonstrated with Configuration C1), computational fluid dynamics (CFD) studies presented a more practical approach for assessing airflow circulation; hence, the CFD models were used for the rest of the study (Configurations C2 and C3). The validated model was used to assess the performance of existing OR configurations using particle circulation as the yardstick—with circulation quantified using PHL and PE as assessment parameters. The existing configurations resulted in PHL and PE as high as 413 s and 3.75 m. High circulation was also observed outside the patient area, which can be attributed to the relatively small outlet duct footprint. Based on the ensuing CFD analyses, an optimal configuration that minimizes circulation was deduced to reduce PHL and maximum PE to 58 s and 1.45 m, respectively.

Adapting PHL and PE as assessment parameters is based on their inherent representation of the exposure time and distance from the inhaling proximity, respectively [40,41]. This is justifiable based on the known heightened transmissibility of airborne aerosols with increased exposure time and closer proximity [41,42]. However, in the OR scenario, the position of the aerosol’s entry routes (nose, eyes, and mouth) varies with the individual height and/or situation (laying, seating, or standing); hence, a single reference height cannot be used to evaluate proximity [43]. Based on this, we consider the inhaling proximity to be higher with particle maximum elevation since, with the floor location of the outlet vents, convection laws enforce subsequent downward circulation towards all elevations below the maximum value. Regardless of the flow type (laminar or turbulent) or ventilation mechanisms (zero or negative pressure), such circulation behavior is expected in ground-located outlet vent systems. However, this does not apply in variable outlet vent elevations; hence, in comparing such systems, a suitable reference point would have to be deduced before adapting PE as an assessment parameter.

The resolved optimized framework, Configuration C3, constitutes a single square inlet duct located on the ceiling above the patient area and four rectangular outlet ducts located in the corners of the room. It is noteworthy that, while the larger outlet duct footprint of Configuration C2 is expected to further reduce circulation, the modular framework of Configuration C3 permits switching to different operation modes depending on the requirement. For instance, since VSs that promote air circulation in the absence of a contamination source are considered healthier, in the absence of power tools, an operation mode with only two adjacent outlet ducts can be adapted. This can then be dynamically switched to an operation mode with all four outlet ducts during periods of surgical smoke or insufflated CO_2_ release to minimize the risk of IAVT.

It is important to note that the primary limitation of theoretical framework of this study lies in the absence of routine surgical activity in the measurements. It is essential to consider the possibility that all members of the surgical team could be in motion or assume different positions within the OR during surgery. For example, in robotic surgery, the surgeon may operate from a considerable distance from the bedside, closer to one of the four corners of the OR. The exposure may vary depending on the position of the surgical robot console. In a recent study, personnel were advised to be aware that moving away from the patient towards a wall in an operating room may prove to be counterproductive [32]. To address this variability, we focused our discussion primarily on regions within the OR rather than personnel, as there are numerous OR setups with different configurations and various surgical team dynamics. Our computer simulation analysis, combined with the experimental airflow validations, confirmed and explained the higher concentration of aerosol particles near the wall, as well as suggesting further OR configurations to minimize the effect for safer ORs. In the future, when these computer simulation studies are combined with experiments to evaluate particle concentration—similar to the work of Wagner et. al. [32]—safer and more enhanced ORs can be designed and implemented.

This study is also limited in that it does not directly compare the effectiveness of the conventional closed systems used for the evacuation of surgery aerosols in laparoscopic surgeries. While the completely closed evacuation systems are intuitively expected to completely eradicate the transmissibility risk, this is not the case for the proposed optimized VS, as particles still circulate within the OR for 58 secs in Configuration C3. In addition, this study only presents a theoretical advantage of Configuration C3 in reducing airborne contamination, as it does not consider the different activities carried out during the surgical phases, the staff size during operations, etc. As such, the VS optimization strategy is recommended as the primary strategy only for scenarios where a completely closed system is not feasible. As such, we envisage the VS optimization strategy to open doors for future improvements to the currently available closed systems, which are not completely closed, with the inevitability of port-site leaks and aerosol escape during operation procedures. Furthermore, such modifications to existing ORs could be costly, and since the VS optimization system is deployed as a general OR protection strategy, its application could permit a reduced reliance on the somewhat tiring and non-user friendly personal protective equipment.

## 5. Conclusions

Currently, evidence on SARS-CoV-2 surgical aerosol transmission remains limited, and there are no studies that evaluate strategies to improve current VS performances for reducing IAVT. Our results suggest that the optimized configuration targeted favorable features from the existing setups and presented a modular framework for a flexible operation mode. While neither a computer-generated spherical particle nor a neutral buoyancy bubble can precisely replicate intraoperative aerosol viral transmission, it is crucial to highlight that the developed computer simulation model and the performed experimental neutral buoyancy bubble tests prioritized capturing key characteristics, particularly mobility, of intraoperative aerosols in ORs. In our simulation model, we incorporated the particle distribution of surgical smoke, which typically ranges from 100 nm to 10.00 μm, with 99.9% of particles collected being <5 μm in size. Notably, we used the average particle size of surgical smoke as the basis for our particle tracing model. The alignment of the neutral buoyancy bubble test results with the conclusions drawn from the simulation model lends support to the notion that neutral buoyancy bubbles reflect the airflow streamlines despite the substantial difference in size compared to aerosols. Therefore, the combination of a computer-based simulation model utilizing particle tracing with the experimental neutral buoyancy bubble tests performed in this study represents a method that comes close to safely characterizing the flow dynamics of aerosol viral transmission in ORs. These findings are clinically important for the following reasons: firstly, they allow us to optimize current VSs and thus be able to use them to their maximum capacity in particular conditions, such as the management of SARS-CoV-2-positive surgical patients. Secondly, the data found in this study may be useful in future pandemics and have the potential to be used as a framework for the development of new guidelines and consensuses, especially during critical moments when the lack of scientific evidence is the most common limitation. Finally, these results will potentially allow us to develop strategies not only to protect the integrity of the patient, but also to improve the safety of the surgical team and healthcare personnel, who are the most exposed during the outbreaks.

## Figures and Tables

**Figure 1 life-14-00313-f001:**
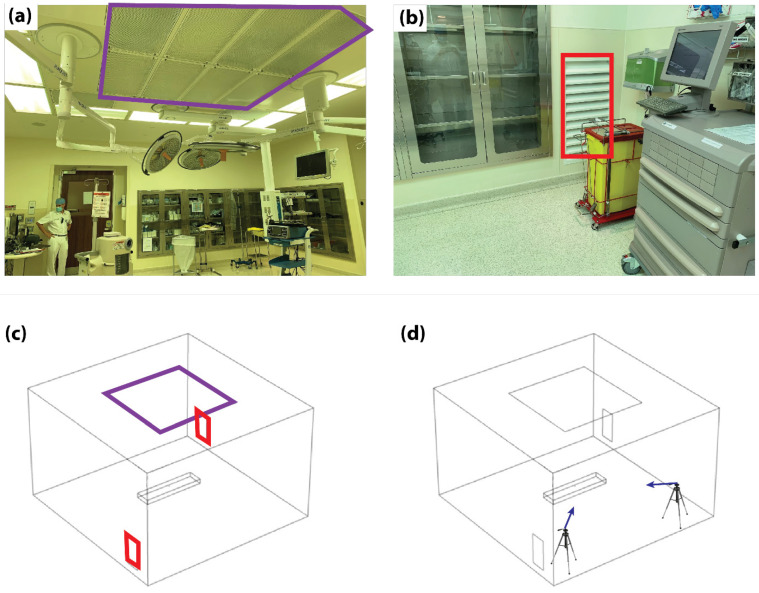
Experimental and numerical setups. (**a**) Image of the real operating room with the inlet duct highlighted (purple square). (**b**) Corner of the operating theater with one of the outlet ducts highlighted (red square). (**c**) Corresponding CFD domain geometry with the inlet (purple square) and outlet ducts (red squares) highlighted. (**d**) Representative schematic highlighting the positions (tripods) and directions (blue arrows) of the particle tracking cameras.

**Figure 2 life-14-00313-f002:**
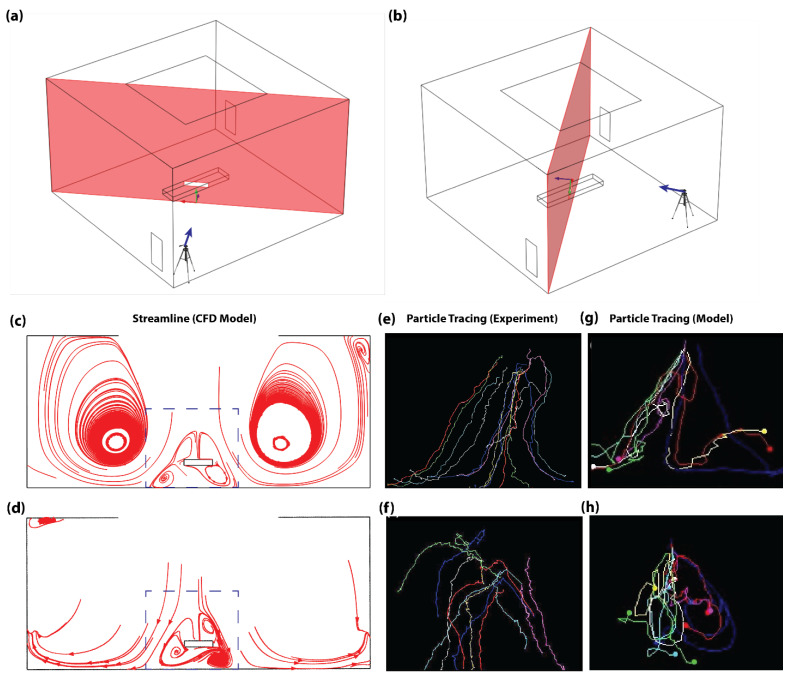
Model validation using measured and CFD-calculated particle streamlines in an empty OR. (**a**) The 3D schematic showing target view plane for the camera setup positioned at the outlet vent corner (View-plane 1). (**b**) The 3D schematic showing target view plane for the camera setup positioned at the corner without an outlet vent (View-plane 2). (**c**) Flow streamline on View-plane 1. (**d**) Flow streamline on View-plane 2. Dotted box in (**c**,**d**) highlights focus area of the camera in experiments. (**e**–**h**) Experimental airflow-tracing bubble tests performed in a fully equipped OR without the presence of the surgical staff. (**e**) Experimentally traced particle trajectory on View-plane 1. (**f**) Experimentally traced particle trajectory on View-plane 2. (**g**) Traced CFD-calculated particle trajectory on View-plane 1. (**h**) Traced CFD-calculated particle trajectory on View-plane 2.

**Figure 3 life-14-00313-f003:**
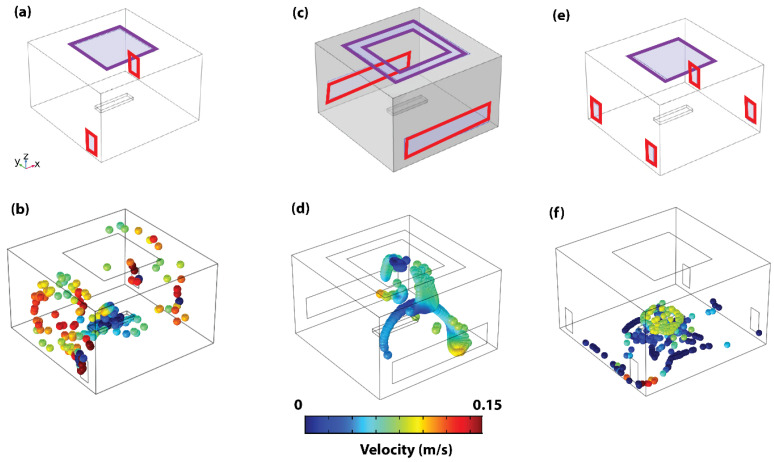
Operation room ventilation system designs and particle dynamics. (**a**) Representative schematic of configuration C1. (**b**) The 3D steady state particle dynamics of configuration C1. (**c**) Representative schematic of configuration C2. (**d**) The 3D particle dynamics of configuration C2. (**e**) Representative schematic of configuration C3. (**f**) The 3D particle dynamics of configuration C3. Inlet ducts highlighted with purple squares and outlet ducts with red rectangles.

**Figure 4 life-14-00313-f004:**
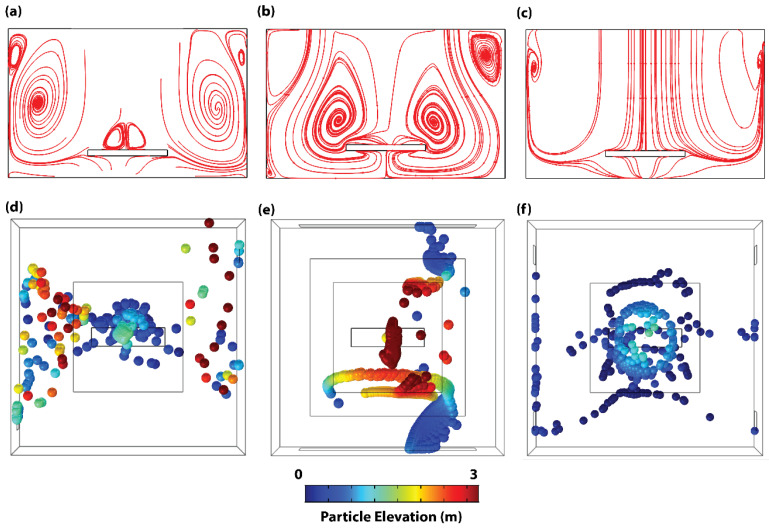
Flow streamline and particle spatial dynamics of adapted configurations. Steady state flow streamlines across the center ZX-plane of (**a**) Configuration C1, (**b**) Configuration C2, and (**c**) Configuration C3. Steady state spatial dynamics of particles in (**d**) Configuration C1, (**e**) Configuration C2, and (**f**) Configuration C3. Particle color map in (**d**–**f**) represents particle elevation.

**Table 1 life-14-00313-t001:** Comparison of the three configurations in terms of PHL and PE.

	Particle Half-Life (s)	Particle Maximum Elevation (m)
Existing Configuration C1	217	3.43
Existing Configuration C2	413	3.75
Proposed Configuration C3	58	1.45

## Data Availability

The datasets generated and/or analyzed during the current study are not publicly available due the nature of the generated CFD models, which are private to the hospital’s operating room specifications, but they are available from the corresponding author on reasonable request.

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
