# Peer review of "Ventilation-Based Strategy to Manage Intraoperative Aerosol Viral Transmission in the Era of SARS-CoV-2"

_life, 2024, doi:10.3390/life14030313_

Round 1

Reviewer 1 Report

Comments and Suggestions for Authors

The authors refer to the "surgical team" (e.g., line 18), by which I think they mean the people doing surgery on the patient at the bedside, compared to those in the operating room. Figure 1b, next to the outlet duct, shows an anesthesia cart. Please, in the revision, consider the impact on the operative team members located near the outlet ducts. See Wagner JA, Dexter F, Greeley DG, Schreiber K. Operating room air delivery design to protect patient and surgical site results in particles released at the surgical table having greater concentration along walls of the room than at the instrument tray. American Journal of Infection Control 49:593-596, 2021. The authors also refer to robotic surgery (e.g., line 282). Consider the implications of whether the primary surgeon would have the console next to one of the four outlet ducts.

Please note that I have not reviewed previously for Life. I do not know the journal. I was the corresponding and principal author of the preceding paper. The editors found this earlier study. Therefore, please compare your results to any other relevant study, the preceding and others published (e.g., its citations [if any] and references). Check your search protocol and why the preceding paper was not identified, and based on modifications assure that any other relevant study is added. Use them to evaluate the limitations of the authors' study, especially generalizability.

The authors released bubbles (line 111). How large are the bubbles relative to the SARS-CoV-2, to which the authors refer? Buoyancy? Relevance? Please address this in the limitations section. I am concerned about the conclusions of "instrumental" (line 328), "clinically important" (line 338), etc.

The authors examined release from "the patient's location" (line 232). However, if I am correct, there was neither a patient nor a surgical team in the OR. In other words, am I correct that the authors studied an empty OR? Then abstract line 21 "flow pattern" sentence should explain "empty OR." The abstract conclusions and paper conclusions both should emphasize empty OR, as should the limitations. Consider adding a study with phantoms, mannequins, or people standing and breathing, to evaluate if the authors' results change.

Author Response

Reviewer #1

The authors refer to the "surgical team" (e.g., line 18), by which I think they mean the people doing surgery on the patient at the bedside, compared to those in the operating room. Figure 1b, next to the outlet duct, shows an anesthesia cart. Please, in the revision, consider the impact on the operative team members located near the outlet ducts. See Wagner JA, Dexter F, Greeley DG, Schreiber K. Operating room air delivery design to protect patient and surgical site results in particles released at the surgical table having greater concentration along walls of the room than at the instrument tray. American Journal of Infection Control 49:593-596, 2021. The authors also refer to robotic surgery (e.g., line 282). Consider the implications of whether the primary surgeon would have the console next to one of the four outlet ducts.

We thank the reviewers for their valuable comment. The surgical team comprises surgeons typically positioned by the patient's bedside, along with operative personnel and anesthesiologists situated at various locations within the operating room (OR). We acknowledge the reviewers’ observation that the exposure of team members may differ based on their typical positions in the OR, and in response, we have incorporated this insight into the discussion by modifying the text as follows:

It is essential to consider the possibility that all members of the surgical team could be in motion or assume different positions within the OR during surgery. For example, in robotic surgery, the surgeon may operate from a considerable distance from the bedside, closer to one of the four corners of the OR. The exposure may vary depending on the position of the surgical robot console. In a recent study, personnel are advised to be aware that moving away from the patient towards a wall in an operating room may prove to be counterproductive. To address this variability, we focused our discussion primarily on regions within the OR rather than personnel, as there are numerous OR setups with different configurations and various surgical team dynamics.

  1. Wagner, J. A., Dexter, F., Greeley, D. G., & Schreiber, K. (2021). Operating room air delivery design to protect patient and surgical site results in particles released at surgical table having greater concentration along walls of the room than at the instrument tray. American journal of infection control, 49(5), 593-596.

Please note that I have not reviewed previously for Life. I do not know the journal. I was the corresponding and principal author of the preceding paper. The editors found this earlier study. Therefore, please compare your results to any other relevant study, the preceding and others published (e.g., its citations [if any] and references). Check your search protocol and why the preceding paper was not identified, and based on modifications assure that any other relevant study is added. Use them to evaluate the limitations of the authors' study, especially generalizability.

We thank the reviewers for the valuable comment. We have added another relevant study that also uses CFD to simulate airflow within various Types of OR ventilation systems and comparing them to class 10 clean room setup. The following text was included in the manuscript and reference as suggested:

More particular, in a recent study, CFD methods were used to simulate airflow within various type of OR setups, suggesting that air exits the space exclusively through a restricted number of outlets—specifically, two to four outlets on just two walls are specified—instead of exiting through a perforated floor, resulting in airflow turbulence in certain areas of the room and unpredictable particle movement.

  1. Wagner JA, P.H.D., Schreiber KJ, Cohen R, P.E. Improving Operating Room Contamination Control. ASHRAE J 2014 02;56(2):18-22,24,26-27.

The authors released bubbles (line 111). How large are the bubbles relative to the SARS-CoV-2, to which the authors refer? Buoyancy? Relevance? Please address this in the limitations section. I am concerned about the conclusions of "instrumental" (line 328), "clinically important" (line 338), etc.

We appreciate the reviewers for posing these questions. In this study, the relevance of the bubbles lies more in their response to the flow field without influencing it, rather than in their size. We have added the following clarification in the limitation section:

The use of near-neutral buoyancy bubbles is not intended to mimic SARS-CoV-2-carrying aerosols, but to dynamically respond to changes in the flow field and to trace flow streamlines, allowing to confirm, the presence of the large vortices generated in the adjacent corners of the OR, as observed in the numerical model.

We also agree with the reviewers regarding their concerns about the use of word “instrumental”, the text is now modified as follows:

We envisage the VS optimization strategy to open doors for future improvements to the currently available closed systems.

The authors examined release from "the patient's location" (line 232). However, if I am correct, there was neither a patient nor a surgical team in the OR. In other words, am I correct that the authors studied an empty OR? Then abstract line 21 "flow pattern" sentence should explain "empty OR." The abstract conclusions and paper conclusions both should emphasize empty OR, as should the limitations. Consider adding a study with phantoms, mannequins, or people standing and breathing, to evaluate if the authors' results change.

We appreciate the reviewers for their valuable comments and suggestions. Few modifications to the text were implemented to emphasis an empty OR. We concur with the reviewers; neither the simulation nor the experimental setup considers the presence of the patient nor the surgical team. The numerical model did not account for the presence of various OR equipment, which, in some instances, may pose greater obstacles than the surgical team. Nevertheless, we found that the experimental observations closely mirror those of the numerical model, particularly regarding the circulation of near-buoyant bubbles in the vortex regions of the OR.

Reviewer 2 Report

Comments and Suggestions for Authors

The topic is intriguing, but it requires revision before it can be recommended for publication.

  1. The literature review in the introduction section is insufficient. While the issues related to the topic are outlined, there is a lack of explanation regarding the solutions, such as operating room design, ventilation system design, and measures to prevent cross-infection. Overall, the authors are encouraged to provide a clear motivation for their work.
  2. A comprehensive review and critique of simulation methods and research advancements is necessary, which the author has not undertaken. The following literature can be referred to:  https://doi.org/10.1016/j.enbuild.2023.113323;  https://doi.org/10.1038/s41467-023-39419-z;  https://doi.org/10.1038/s41586-020-2271-3;  https://doi.org/10.1038/s41598-021-91265-5.
  3. The author should offer sufficient explanation and justification for the advantages and effectiveness of the chosen methods. What are the advantages of this simulation design? The author should address and enhance this aspect.
  4. Has the author conducted appropriate or similar experimental comparisons for their simulation calculations? What recommendations can be made to improve existing ventilation systems and operating theatres? The author should emphasize this point.

Minor suggestions:

  1. Separate the conclusion section.
  2. There are formatting issues in the PDF file.

Author Response

Reviewer #2

The topic is intriguing, but it requires revision before it can be recommended for publication.

  • The literature review in the introduction section is insufficient. While the issues related to the topic are outlined, there is a lack of explanation regarding the solutions, such as operating room design, ventilation system design, and measures to prevent cross-infection. Overall, the authors are encouraged to provide a clear motivation for their work.

We thank the reviewers for their valuable suggestion. Indeed, providing more explanation regarding the solution would strengthen the introduction. In this regard, we have incorporated the following text:

In modern operating room air distribution design, primary considerations revolve around maintaining sterile surgical zone and integration with medical equipment.  On one hand, the objective is to establish and uphold a sterile surgical zone encompassing both the patient and the surgical team. On the other, the design must seamlessly coordinate with sophisticated medical imaging devices, task lighting, medical gas columns, and other ceiling-mounted equipment, ensuring optimal functionality within the operating room. The guidelines provided by FGI (Facility Guidelines Institute) for the Construction of Health Care Facilities and the ASHRAE 170-2017 Standard for the Ventilation of Health Care Facilities delineate specific airflow requirements for operating rooms. For instance, the room should maintain positive pressurization, with a minimum of 20 total air changes per hour (ACH). Additionally, the airflow, filtered appropriately, should follow a unidirectional downward pattern through a laminar array designed for 25 to 35 cfm/sq.ft. Furthermore, a primary supply diffuser array must cover at least 70% of the surgical zone, defined by the footprint of the operating table and a 12-inch perimeter offset. In the context of traditional operating rooms, there is typically an ample space to accommodate a large laminar diffuser array directly above the operating table. This spatial arrangement facilitates the straightforward fulfillment of the specified ASHRAE 170 criteria when devising a ventilation and ceiling system for optimal operating room functionality.

  1. Wagner JA, P.H.D., Schreiber KJ, Cohen R, P.E. Improving Operating Room Contamination Control. ASHRAE J 2014 02;56(2):18-22,24,26-27.

  • A comprehensive review and critique of simulation methods and research advancements is necessary, which the author has not undertaken. The following literature can be referred to: https://doi.org/10.1016/j.enbuild.2023.113323;  https://doi.org/10.1038/s41467-023-39419-z;  https://doi.org/10.1038/s41586-020-2271-3;  https://doi.org/10.1038/s41598-021-91265-5.

We thank the reviewers for their valuable comment. Indeed, information provided in these suggested references would add improvement to the manuscript. We have looked into the suggested literature and references’ relevant material, and added the following text:

In the past 3 decades, several studies have contributed to the development of effective guidelines for controlling airborne infections in healthcare settings. One example of such guidelines is the requirement to maintain a minimum air change rate of 20 per hour, in the operating room (OR), to minimize the deposition of airborne contaminants in the surgical area and reduce the likelihood of surgical patients contracting surgical site infections. In another relevant study, an overview of models and algorithms utilized in Computational Fluid Dynamics (CFD) simulations for pathogen transmission research is provided. This work highlights the representation of pathogens as particles or gases, integrates epidemic models for enhanced accuracy, and emphasizes the crucial role of airflow patterns in pathogen transmission. Additionally, the study introduces advanced CFD methods such as the Lattice Boltzmann Method (LBM), Porous Media models, and Web-based approaches, offering a diverse set of tools for intricate investigations in this field. More particular, in a recent study, CFD methods were used to simulate airflow within various type of OR setups, suggesting that air exits the space exclusively through a restricted number of outlets—specifically, two to four outlets on just two walls are specified—instead of exiting through a perforated floor, resulting in airflow turbulence in certain areas of the room and unpredictable particle movement.

  1. Kek, H. Y., Saupi, S. B. M., Tan, H., Othman, M. H. D., Nyakuma, B. B., Goh, P. S., ... & Wong, K. Y. (2023). Ventilation Strategies for Mitigating Airborne Infection in Healthcare Facilities: A Review and Bibliometric Analysis (1993 to 2022). Energy and Buildings, 113323.
  2. Puthussery, J.V., Ghumra, D.P., McBrearty, K.R. et al. Real-time environmental surveillance of SARS-CoV-2 aerosols. Nat Commun 14, 3692 (2023). https://doi.org/10.1038/s41467-023-39419-z
  3. Carducci, A., Federigi, I., & Verani, M. (2020). Covid-19 airborne transmission and its prevention: waiting for evidence or applying the precautionary principle?. Atmosphere, 11(7), 710.
  4. Wagner JA, P.H.D., Schreiber KJ, Cohen R, P.E. Improving Operating Room Contamination Control. ASHRAE J 2014 02;56(2):18-22,24,26-27.
  5. Peng, S., Chen, Q., & Liu, E. (2020). The role of computational fluid dynamics tools on investigation of pathogen transmission: Prevention and control. Science of The Total Environment, 746, 142090.
  • The author should offer sufficient explanation and justification for the advantages and effectiveness of the chosen methods. What are the advantages of this simulation design? The author should address and enhance this aspect.

We appreciate the reviewers for their valuable suggestions. Addressing the reviewers’ comment, we have added the following explanation and justification to the text:

The techniques employed to examine and assess the effectiveness of operating room designs concerning particle movement within and around the sterile field include: 1) Computational fluid dynamic modeling (CFD) and 2) Airflow simulation using neutral buoyancy bubbles. CFD has being traditionally used to illustrate how airflow patterns within the OR can affect the induction and migration of particles. In a recent study, particle trace analysis comparing the Moving Downstream Air (MDA) design with Single Laminar Diffuser (SLD) design reveals that SLD performs significantly better in pulling contaminants away from the surgical site under equivalent conditions. Additionally, neutral buoyancy bubbles experiment reveals the impact of low-pressure zones resulting from airflow blockage.

  1. Wagner, J.A., and K.J. Schreiber. “Comparing air delivery methods in an operating theater: cleanroom technology can improve contaminant control and removal.” Submitted December 2013, HVAC&R Research.
  • Has the author conducted appropriate or similar experimental comparisons for their simulation calculations? What recommendations can be made to improve existing ventilation systems and operating theatres? The author should emphasize this point.

We appreciate the reviewers for their comment. Indeed, the techniques we deployed for this study are computational (CFD) and experimental using neutral buoyancy bubbles. Both techniques were compared (253-256). And recommendations have been made to improve existing ventilation systems (273-280). In order to further emphasize these points, the following text is added:

In this updated configuration, we suggest employing a square outlet duct and four inlet ducts positioned at the corners of the room (refer to Figure 3e). The intention behind covering all four corners of the operating room is to minimize any negative pressure spots that might alter airflow patterns, potentially surpassing our designated particle maximum elevation threshold.

Minor suggestions:

  • Separate the conclusion section.

We agree with the reviewers on separating the conclusion section, the change has been implemented in the text.

  • There are formatting issues in the PDF file.

We appreciate the reviewers for their minor suggestions we have solve all formatting issues in the PDF file.

Reviewer 3 Report

Comments and Suggestions for Authors

Regarding the manuscript entitled: "Ventilation based strategy to manage intraoperative aerosol viral transmission in the era of SARS-CoV-2”, I would like to congratulate the authors for this article that may be relevant in optimizing operating room ventilation systems to minimize aerosol exposure of surgical personnel.

Introduction: Adequate.

I believe that the main strength of the study is its potential application after what happened in the COVID19 pandemic. Structural modification of operating rooms can prevent surgical procedures from being suspended in a potential future pandemic, with the devastating consequences it may have for patients. In this sense, the structural implementation of the operating rooms would be in line with the processes of guarantees of non-repetition proclaimed by the application of Transitional Justice (DOI: 10.3390/ijerph191912388). This information could be added as a justification for the research on the avoidance of the spread of airborne particles and the consequent spread of SARS-CoV-2.

Material and methods:

I consider that the authors should present at the beginning of this section the type of study carried out in a generic way before entering into the presentation of the methodology.

Likewise, it must be established if it has been registered in a trial database or if the protocol has been approved by an ethics committee or verified by a university committee. Although the study does not involve experimentation on patients, this step seems essential to increase credibility in the adequacy of the methodology.

The meaning of the acronyms CFD, DSLRS and GMRES should be explained before using them.

I think that the different configurations expressed in the text (C1, C2 and C3), theoretically figures 3a, 3b and 3c (lines: 172 – 181), are incorrectly indicated in figure 3 (I think they are 3a, 3c and 3e). I think the order of the figures needs to be changed

Results:

The first sentence of this section has already been explained in the material and methods section, so it could be deleted (lines 199-201).

In my opinion, the results section could be summarized so that the findings detected in this research are clearer to potential readers.

In the paragraph contained between lines 223-238, the presentation of results is mixed with the "explanation of the potential cause that may have led to these results." In my opinion this can be confusing and make the results detected less relevant. I recommend taking those "assumptions (can be attributed)" or "explanation" of the results to the discussion section.

Furthermore, the information about the explanation of each of the configurations should be made in the material and methods section. For example: information provided in lines 240-241 "which consists of a square... at the corners of the room".

Discussion

I think it should be stated in the limitations section that the main limitation of this study is its theoretical conception and the non-involvement of routine surgical activity in the measurements.

Author Response

Reviewer #3

  • Regarding the manuscript entitled: "Ventilation based strategy to manage intraoperative aerosol viral transmission in the era of SARS-CoV-2”, I would like to congratulate the authors for this article that may be relevant in optimizing operating room ventilation systems to minimize aerosol exposure of surgical personnel.

We would like to express our gratitude for the reviewers’ kind words and interest in our work. Our intention is indeed to pave the way for further exploration of optimizing the operating room ventilation system to minimize the exposure of surgical personnel. We value the reviewers’ comments and will thoroughly peruse through them to further improve the quality of the manuscript.

            Introduction: Adequate.

  • I believe that the main strength of the study is its potential application after what happened in the COVID19 pandemic. Structural modification of operating rooms can prevent surgical procedures from being suspended in a potential future pandemic, with the devastating consequences it may have for patients. In this sense, the structural implementation of the operating rooms would be in line with the processes of guarantees of non-repetition proclaimed by the application of Transitional Justice (DOI: 10.3390/ijerph191912388). This information could be added as a justification for the research on the avoidance of the spread of airborne particles and the consequent spread of SARS-CoV-2.

We appreciate the reviewers for their comment. Indeed, any changes or modifications to the structure in operating rooms has the potential to avert the suspension of surgical procedures during a possible future pandemic, thereby avoiding the severe consequences it could inflict on patients and the surgical team. In this context, we have added the suggested justification to the text as follows:

Any changes or modifications to the structure in operating rooms has the potential to avert the suspension of surgical procedures during a possible future pandemic, thereby avoiding the severe consequences it could inflict on patients and the surgical team. In this regard the structural implementation of the operating rooms would be in line with the processes of guarantees of non-repetition proclaimed by the application of Transitional Justice.

  1. Rodríguez Reveggino, B.; Becerra-Bolaños, Á. Transitional Justice after the COVID-19 Pandemic. Int. J. Environ. Res. Public Health 2022, 19, 12388. https://doi.org/10.3390/ijerph191912388

Material and methods:

  • I consider that the authors should present at the beginning of this section the type of study carried out in a generic way before entering into the presentation of the methodology.

We appreciate and agree with the reviewers for their suggestion. To this end, we have added the following text at the beginning of the section:

The techniques employed to examine and assess the effectiveness of operating room designs concerning particle movement within and around the sterile field include: 1) Computational fluid dynamic modeling (CFD) and 2) Airflow simulation using neutral buoyancy bubbles. CFD has being traditionally used to illustrate how airflow patterns within the OR can affect the induction and migration of particles. In a recent study, particle trace analysis comparing the Moving Downstream Air (MDA) design with Single Laminar Diffuser (SLD) design reveals that SLD performs significantly better in pulling contaminants away from the surgical site under equivalent conditions. Additionally, neutral buoyancy bubbles experiment reveals the impact of low-pressure zones resulting from airflow blockage.

  1. Wagner, J.A., and K.J. Schreiber. “Comparing air delivery methods in an operating theater: cleanroom technology can improve contaminant control and removal.” Submitted December 2013, HVAC&R Research.
  • Likewise, it must be established if it has been registered in a trial database or if the protocol has been approved by an ethics committee or verified by a university committee. Although the study does not involve experimentation on patients, this step seems essential to increase credibility in the adequacy of the methodology.

We appreciate the reviewer's comment. The protocol in question did not undergo ethics committee approval as it did not involve patients. However, our future work intends to expand on this study by incorporating patients and the entire surgical team in the operating room (OR). We acknowledge the importance of ethical considerations and plan to seek approval from the relevant Institutional Review Board (IRB) committees associated with the hospital for our upcoming protocols.

  • The meaning of the acronyms CFD, DSLRS and GMRES should be explained before using them.

We appreciate the reviewers for pointing out these missed acronyms. The text has been now modified to provide the meaning for all acronyms in the manuscript.

  • I think that the different configurations expressed in the text (C1, C2 and C3), theoretically figures 3a, 3b and 3c (lines: 172 – 181), are incorrectly indicated in figure 3 (I think they are 3a, 3c and 3e). I think the order of the figures needs to be changed

We thank the reviewers for pointing out this typo in the text. Corrections have been now made to the manuscript in this regard.

Results:

  • The first sentence of this section has already been explained in the material and methods section, so it could be deleted (lines 199-201). In my opinion, the results section could be summarized so that the findings detected in this research are clearer to potential readers.

We appreciate the reviewers for their suggestion. Accordingly, we have omitted the first sentence of the result section.

  • In the paragraph contained between lines 223-238, the presentation of results is mixed with the "explanation of the potential cause that may have led to these results." In my opinion this can be confusing and make the results detected less relevant. I recommend taking those "assumptions (can be attributed)" or "explanation" of the results to the discussion section.

We thank the reviewers for their suggestion. Indeed, language indicating a sense of assumption or explanation of the results should be kept for the discussion section. However, we believe that all the information provided in this paragraph are results observations that strengthen the results section. To this effect, few modifications are added to the text as follows:

The significant increase in PHL is observed to occur as a result of repeated circulation within the patient’s location.

  • Furthermore, the information about the explanation of each of the configurations should be made in the material and methods section. For example: information provided in lines 240-241 "which consists of a square... at the corners of the room".

We appreciate the reviewers for their comment. Changes have been implemented to the text accordingly.

Discussion

  • I think it should be stated in the limitations section that the main limitation of this study is its theoretical conception and the non-involvement of routine surgical activity in the measurements.

We thank the reviewers for pointing out this important point. We believe we have mentioned this point in the discussion section as follows:

In addition, this study only presents a theoretical advantage of Configuration C3 in reducing airborne contamination as it does not consider the different activities carried out during the surgical phases, the staff size during operations, etc.

We have also now emphasized it further in the beginning of the paragraph as follows:

It is important to note, that the primary limitation of theoretical framework of this study lies in the absence of routine surgical activity in the measurements. It is essential to consider the possibility that all members of the surgical team could be in motion or assume different positions within the OR during surgery. For example, in robotic surgery, the surgeon may operate from a considerable distance from the bedside, closer to one of the four corners of the OR. The exposure may vary depending on the position of the surgical robot console. To address this variability, we focused our discussion primarily on regions within the OR rather than personnel, as there are numerous OR setups with different configurations.

Round 2

Reviewer 1 Report

Comments and Suggestions for Authors

My concerns are solely minor issues of writing.

Revise the title line 2 and conclusions lines 403 and 409 because the authors have neither used computer simulation nor buoyancy study to examine "aerosol viral transmission." In the response, they explain that neither represents the movement of SARS-CoV-2 in an operating room.

The presentation should clarify whether the experiments were performed in an empty operating room or an operating room without people (i.e., Figure 1 picture b). Clarify in the Figure 1 legend, either way. Use the phrase "without people" (or equivalent phrase) or "empty" at lines 128, 170, 329, and 405.

In my opinion, the use of Reference 32 has been incomplete. As I wrote in my earlier review, I do not suggest that this one article is complete. The authors did not address the literature search about the evidence of the importance of register location, and that would be the location for SARS-CoV-2, although it is logical. The novel feature of the authors' results clinically focuses on the registers. For example, suppose there have yet to be additional experimental studies. In the Introduction, explain that the earlier work is limited as having been experimental without simulation or understanding mechanism. The authors' study rectifies that limitation. Alternatively, or in addition, in the discussion, add the findings as further evidence of the validity of the authors' results. (If it does not add validity, say that, but I think it does.) The authors studied either an empty OR or one with full equipment but without people. Another study with matching conclusions provides greater support for the authors' conclusions.

Author Response

We greatly appreciate the reviewer’s thoughtful comments and the valuable suggestions to improve our manuscript. Below are the reviewer’s comments (in black), our responses (in blue), and quotes from the revised manuscript (in red).

My concerns are solely minor issues of writing.

  • Revise the title line 2 and conclusions lines 403 and 409 because the authors have neither used computer simulation nor buoyancy study to examine "aerosol viral transmission." In the response, they explain that neither represents the movement of SARS-CoV-2 in an operating room.

We appreciate the feedback provided by the reviewer. The title does not suggest that the manuscript is simulating aerosol SARS-Cov-2 movement in an operating room. With all due respect to the reviewer, the title is very clear by stating this work as “Ventilation Based Strategy to Manage Intraoperative Aerosol Viral Transmission” “in the Era of SARS-CoV-2”. The study was motivated by the pandemic of COVID-19, and experiments were conducted in OR amid the pandemic. The motivation of this work is to study airflow dynamics in OR, and Intraoperative Aerosol Viral Transmission is well established in the literature as to be carried with the airflow in the OR. Therefore, we believe the current title of the paper represent the work well. However, and to address the reviewer’s concern, we added the following paragraph to the conclusions section of the manuscript:

While neither a computer-generated spherical particle nor a neutral buoyancy bubble can precisely replicate the Intraoperative Aerosol Viral Transmission, it's crucial to highlight that the developed computer simulation model and the performed experimental neutral buoyancy bubble tests prioritized capturing key characteristics, particularly mobility, of Intraoperative Aerosols in ORs. In our simulation model, we incorporated the particle distribution of surgical smoke, which typically ranges from 100 nm to 10.00 μm, with 99.9% of particles collected being <5 μm in size. Notably, we used the average particle size of surgical smoke as the basis for our particle tracing model. The alignment of the neutral buoyancy bubble test results with the conclusions drawn from the simulation model lends support to the notion that neutral buoyancy bubbles reflect the airflow streamlines, despite the substantial difference in size compared to aerosols. Therefore, the combination of computer-based simulation model utilizing particle tracing with the experimental neutral buoyancy bubble tests performed in this study represents a method that comes close to safely characterizing such flow dynamics of Aerosol Viral Transmission in ORs.

  • The presentation should clarify whether the experiments were performed in an empty operating room or an operating room without people (i.e., Figure 1 picture b). Clarify in the Figure 1 legend, either way. Use the phrase "without people" (or equivalent phrase) or "empty" at lines 128, 170, 329, and 405.

We appreciate the reviewer for the comment. This concern has been addressed in our previous revisions, and we have incorporated the term "Empty OR" in lines 138 and 139 to specifically respond to this comment in the earlier version. However, and to address the reviewer’s continuing concern, we performed the following edits to the manuscript:

  • Lines 137 to 144: In this study, we used the commercially available multiphysics computer-based simulation package, COMSOL, to develop CFD models of a standard empty OR with varying ventilation configurations. Furthermore, experimental airflow patterns collected from an existing fully-equipped OR were compared with the model’s result for validations. The experimental airflow patterns were measured by tracing the flow of bubbles in a fully-equipped OR, but without the presence of the surgical team. In view of minimizing surgical aerosols circulation, the airflow pattern produced from three different ventilation configurations were evaluated using particle half-life (PHL) and particle elevation (PE).

  • As Figure 2 is first to explain the results, we edited it’s caption as the following: Figure 2. Model validation using measured and CFD calculated particle streamlines in an empty OR. (a) 3D schematic showing target view plane for the camera setup positioned at the outlet vent corner (View-plane 1). (b) 3D schematic showing target view plane for the camera setup positioned at the corner without an outlet vent (View-plane 2). (c) Flow streamline on View-plane 1. (d) Flow streamline on View-plane 2. Dotted box in (c) & (d) highlights focus area of the camera in experiments. (e-h) Experimental airflow-tracing bubble tests performed in a fully-equipped OR without the presence of the surgical staff. (e) Experimentally traced particle trajectory on View-plane 1. (f) Experimentally traced particle trajectory on View-plane 2. (g) Traced CFD calculated particle trajectory on View-plane 1. (h) Traced CFD calculated particle trajectory on View-plane 2.

  • Lines 336-344: In this study, the particle tracing numerical model was developed for the configuration of an existing and functional OR, and representative particle tracing experiments were used to validate the computational model. It is important to note here that the computer numeral model represented an empty OR, while the experimental tracing tests were performed in a fully equipped OR but without the presence of the surgical team. The use of near-neutral buoyancy bubbles is not intended to mimic SARS-CoV-2-carrying aerosols, but to dynamically respond to changes in the flow field and to trace flow streamlines, allowing to confirm, the presence of the large vortices generated in the adjacent corners of the OR, as observed in the numerical model.
  • In my opinion, the use of Reference 32 has been incomplete. As I wrote in my earlier review, I do not suggest that this one article is complete. The authors did not address the literature search about the evidence of the importance of register location, and that would be the location for SARS-CoV-2, although it is logical. The novel feature of the authors' results clinically focuses on the registers. For example, suppose there have yet to be additional experimental studies. In the Introduction, explain that the earlier work is limited as having been experimental without simulation or understanding mechanism. The authors' study rectifies that limitation. Alternatively, or in addition, in the discussion, add the findings as further evidence of the validity of the authors' results. (If it does not add validity, say that, but I think it does.) The authors studied either an empty OR or one with full equipment but without people. Another study with matching conclusions provides greater support for the authors' conclusions.

We appreciate the feedback provided by the reviewer. To address this valuable recommendation, we added the following paragraphs:

  • In the Introduction section: A recent study by Wagner et. al.32 performed experimental tests to evaluate air particle concentrations in ORs. The tests were done by releasing particles at the OR table and consequently evaluating particle concentration within the room. Their data showed that the airflow within the OR caused higher particle concentration neat the OR walls in comparison to the centre of the room. While this promise for a safer zone for patients and surgeons, the rest of the surgical team including surgical assistants, nurses, anaesthetists, and other operators might face higher risks. The study did not explain what are the risks involved in this particle concentration near the walls and did not explain how to solve the problem.

  • Lines 301-303: These findings explain the higher particle concentration near the OR’s walls reported earlier32. This confirms the power of combining computer simulation-based analysis with experimental tests.

  • Lines 401-406: Our computer simulation analysis, combined with the experimental airflow validations, confirmed and explained the higher concentration of aerosol particles near the wall, and further suggested OR configurations to minimize the effect for safer ORs. In the future, when these computer simulation studies are combined with experiments to evaluate particle concentration – similar to the work of Wagner et. al.32 – safer and more enhanced ORs can be designed and implemented.

Reviewer 2 Report

Comments and Suggestions for Authors

I am satisfied with the author's revision and response on this occasion. I wholeheartedly recommend that this revision be accepted for publication.

Author Response

Dear Reviewer,

Thank you for your kind words. We are pleased to hear that you are satisfied with the revised version of the manuscript and that you recommend it for publication.

We greatly appreciate your feedback and recommendations.

Best regards,

The Authors